# Global emergence of double and multi-carbapenemase producing organisms: epidemiology, clinical significance, and evolutionary benefits on antimicrobial resistance and virulence

Pei-Bo Yuan,[1] Li-Ting Dai,[1] Qi-Ke Zhang,[1] Yu-Xia Zhong,[1] Wan-Ting Liu,[1] Ling Yang,[2] Ding-Qiang Chen[1]

**ABSTRACT** Redundant carbapenemase-producing (RCP) bacteria, which carry double or multiple carbapenemases, represent a new and concerning phenomenon. The objective of this study is to conduct a comprehensive analysis of the epidemiology and genetic mechanisms of RCP strains to support targeted surveillance and control measures. A retrospective analysis was conducted using surveillance data from 277 articles. Statistical analysis was performed to determine and evaluate species prevalence, proportions of carbapenemases, antibiotic susceptibility profiles, sample information, and patient outcomes. Complete plasmid sequencing data were utilized to investigate potential antimicrobial resistance or virulence advantages that strains may gain from acquiring redundant carbapenemases. RCP bacteria are widely distributed globally, and their prevalence is increasing over time. Several countries, including China, India, Iran, Turkey, and South Korea, have reported more than 100 RCP strains. The most commonly reported RCP species are *Klebsiella pneumoniae* and *Acinetobacter baumannii*, which exhibit varying proportions of carbapenemase combinations. Certain species-carbapenemase combinations, such as *K. pneumoniae* carrying New Delhi metallo-β-lactamase (NDM) + oxacillinase (OXA) (56.76%) and *K. pneumoniae* carbapenemase (KPC) + Verona integron-encoded metallo-β-lactamase (VIM) (50.00%) carbapenemases, are associated with high mortality rates. In patients with RCP strains isolated from the bloodstream and respiratory system, the mortality rates are 58.70% and 69.23%, respectively. Analysis of plasmids from RCP strains suggests that they may acquire additional antibiotic resistance phenotypes and virulence factors. Carbapenem-resistant bacteria carrying redundant carbapenemases pose a significant global health threat. This study provides valuable insights into the epidemiology and genetic mechanisms of these bacteria, supporting the development of effective control and prevention strategies to mitigate their transmission.

**IMPORTANCE** This study examined the global distribution patterns of 1,780 bacteria with double or multiple carbapenemases from 277 articles and assessed their clinical impact. The presence of multiple carbapenemases increases the chances of co-resistance to other classes of antibiotics and more virulence factors, further complicating the clinical management of infections.

**KEYWORDS** redundant carbapenemases-producing (RCP) bacteria, epidemiology, clinical significance, bacterial benefits, carbapenem resistance

Address correspondence to Ding-Qiang Chen, jyksys@126.com.

The authors declare no conflict of interest.

See the funding table on p. 11.

C arbapenems are effective antibiotics used to treat severe bacterial infections. These antimicrobials are often reserved as a last resort for treating infections caused by multidrug-resistant pathogens (1). However, the emergence and dissemination of carbapenem-resistant organisms (CROs) have greatly limited the efficacy of these

antibiotics (2, 3). Carbapenem resistance can arise through several mechanisms, one common mechanism is the production of carbapenemases, which are enzymes that cleave carbapenem's β-lactam ring and render the antibiotic ineffective (4). These carbapenemases are often encoded by mobile genetic elements, which can spread rapidly within a bacterial population and between bacterial species, leading to the dissemination of carbapenem resistance (5).

There are several classes of carbapenemases, including *Klebsiella pneumoniae* carbapenemase (KPC), New Delhi metallo-β-lactamase (NDM), Imipenemase metallo-β-lactamase (IMP), Verona integron-encoded metallo-β-lactamase (VIM), Oxacillinase (OXA), and Guiana extended-spectrum β-lactamase (GES) (5). These carbapenemases differ in their biochemical properties, substrate specificity, and mechanisms of action (6). They also vary in their prevalence and geographic distribution, depending on the bacterial species and their resistance mechanisms (7). Due to the variations in hydrolytic profiles and inhibitor susceptibility of different carbapenemase enzymes, differentiated treatment strategies have been developed in clinical practice to target different carbapenemases (3, 8). For example, ceftazidime-avibactam and meropenem-vaborbactam have been found to effectively against KPC-producing Enterobacterales, but show limited efficacy against metallo beta lactamase (MBL)-producing bacteria (including NDM, IMP, and VIM). Most VIM-producing isolates are susceptible to cefepime-tazobactam due to the weak activity of VIM against cefepime specifically, but KPC and NDM-producing Enterobacterales are mostly resistant (9). Aztreonam is not hydrolyzed by carbapenemase NDM and therefore active against isolated NDM producers. However, organisms that exhibit multiple resistance mechanisms (e.g., ESBL, other carbapenemases, and porin/efflux mutations) can result in the inactivation of aztreonam (3, 10).

The emergence of carbapenem-resistant bacterial strains carrying redundant carbapenemases is a noteworthy new phenomenon (11). Simultaneous carriage of more than one carbapenemase can potentially provide carbapenem-resistant strains with higher hydrolytic capabilities to carbapenem and a broader antibiotic resistance profile, further limiting the already restricted choices for effective antibiotics.

Understanding the geographical distribution, temporal trends, species prevalence, enzyme types, clinical sample sources, and patient outcomes associated with these strains is crucial to developing effective strategies for combating this novel antibiotic resistance threat. This study aimed to gain a comprehensive understanding of the characteristics, distribution, clinical impact, and possible bacterial benefits of redundant carbapenemases-producing (RCP) strains, aiding in the development of effective strategies to detect and combat antibiotic resistance.

## RESULTS

### Global distribution of carbapenem-resistant bacteria producing redundant carbapenemases

A total of 1,780 RCP strains have been reported across multiple continents, including 1,179 strains from Asia, 294 strains from Africa, 275 strains from Europe, 87 strains from the Americas, and 6 strains from Oceania. These findings are based on data from 277 articles, and more detailed information can be found in Table S2. Among the countries, China has reported the highest number of RCP isolates, followed by India, Iran, Thailand, Turkey, and South Korea, all of which have reported more than 100 isolates (Fig. 1A).

The main species carrying redundant carbapenemases include *Klebsiella* spp., *Acinetobacter* spp., *Citrobacter* spp., *Enterobacter* spp., *Escherichia coli*, and *Pseudomonas aeruginosa*, etc. However, the distribution of these species varies among continents (Fig. 1B). A total of 28 different carbapenemase combinations have been reported, with the top five combinations being NDM coexistence with OXA, KPC coexistence with NDM, VIM coexistence with OXA, KPC coexistence with VIM, and KPC coexistence with OXA, accounting for approximately 70% of the total number of combinations reported across all continents (Fig. 1C).

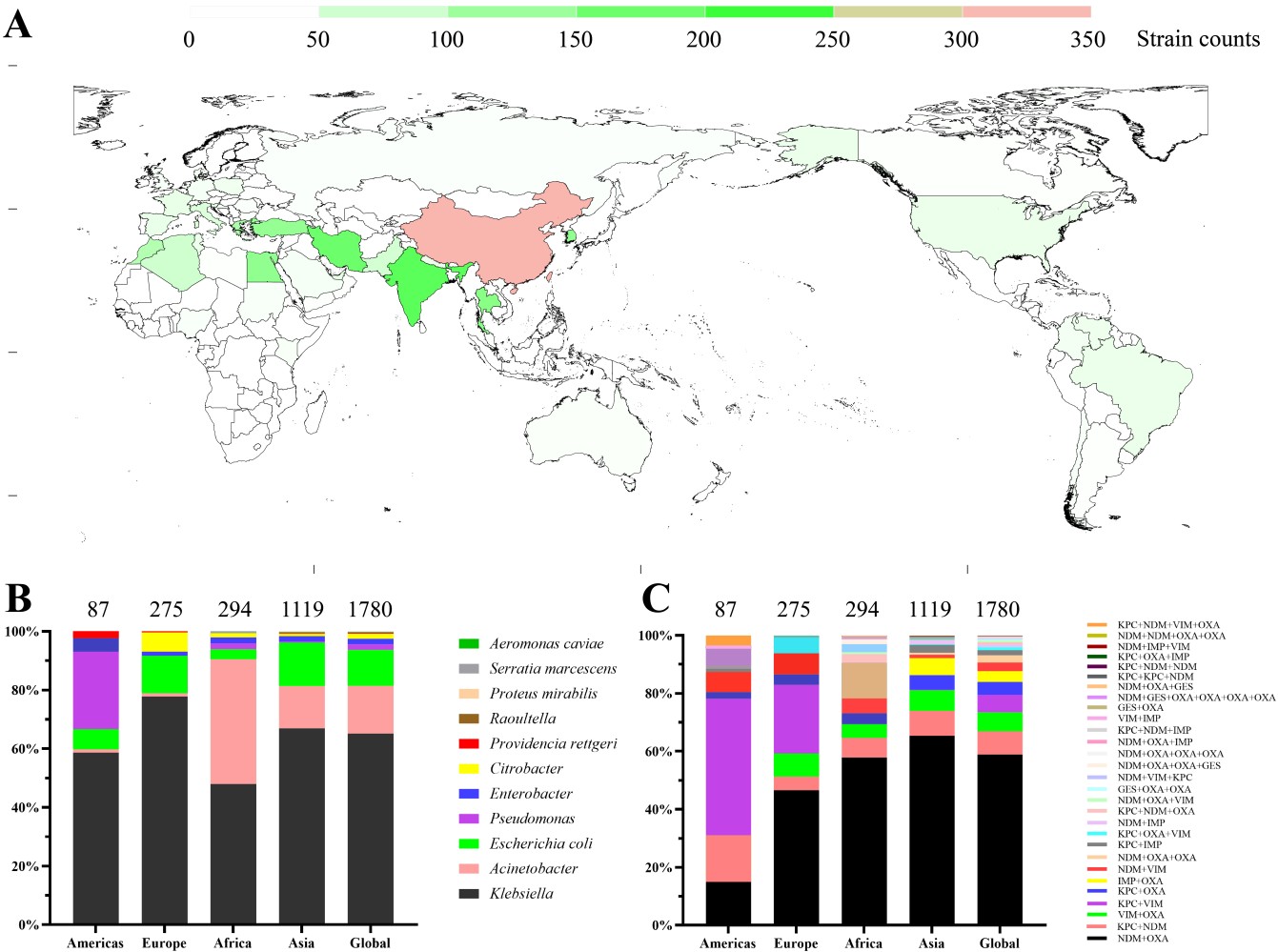

**FIG 1** Geographic distribution of RCP strains. (A) Epidemiological features of RCP strains by country of origin. The quantities of reported strains in different regions are visualized using a color-coded scheme. (B) Species distribution of RCP strains among continents. (C) Carbapenemase combination distribution of RCP strains among continents.

## The temporal trend and species-carbapenemase proportion of RCP strains

The carbapenemase combinations were further classified based on the number of carbapenemases (Fig. 2A). Double-carbapenemase-producing (DCP) strains were the most prevalent, accounting for 93.48% of all isolates. There were 111 triple-carbapenemase-producing (TCP) strains, eight quadruple-carbapenemase-producing (QCP) strains, and only one sextuple-carbapenemase-producing (SCP) strain. To examine the temporal patterns and fluctuations in the prevalence of strains producing double or multi-carbapenemases, the publication years of relevant studies were analyzed (Fig. 2B). An increasing number of RCP strains have been reported over time. The number of DCP strains began to increase in 2009 and remained consistently above 150 strains per year since 2016. The first report of TCP strains dates back to 2011, reaching its peak in 2017 with 36 reported strains. QCP strains have been reported in recent years, specifically in 2019 ($n = 3$), 2020 ($n = 3$), and 2022 ($n = 2$). Additionally, a single case of an SCP strain was reported in Egypt in 2020, producing NDM-25, OXA-23, OXA-48, OXA-51, OXA-181, and GES-1 (12).

To investigate the association between different carbapenemase combinations and bacterial species, we examined the distribution of species and carbapenemase proportions in DCP strains (Fig. 2C) and TCP strains (Fig. 2D). Among 1,664 DCP isolates,

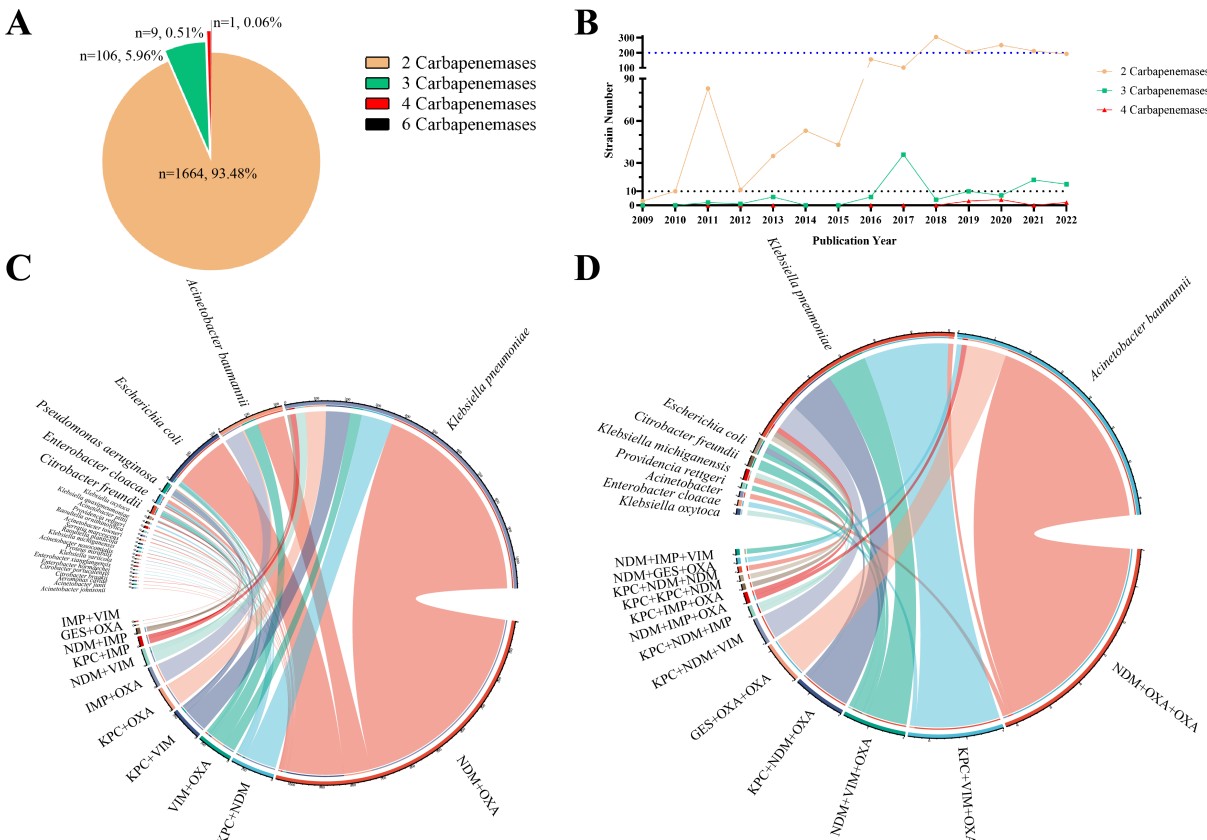

**FIG 2** Temporal trend and molecular epidemiology of RCP strains. (A) The proportion of strains producing carbapenemases is categorized by the number of carbapenemases present. (B) Temporal trend of RCP strains. (C) Distribution of species-carbapenemase combinations in strains producing two carbapenemases. (D) Distribution of species-carbapenemase combinations in strains producing three carbapenemases.

*K. pneumoniae* was the dominant species (65.75%, 1,094/1,664), producing NDM+OXA (66.45%), KPC+NDM (10.05%), KPC+VIM (7.59%), and KPC+OXA (6.49%) as the leading carbapenemase combinations. The second most prevalent species was *Acinetobacter baumannii* (13.16%, 219/1,664), followed by *E. coli* (12.92%, 215/1,664), with over 200 DCP strains reported for both species. The leading carbapenemase combinations for DCP *A. baumannii* were NDM+OXA (43.38%), IMP+OXA (30.15%), and VIM+OXA (22.83%), while for DCP *E. coli*, they were NDM+OXA (90.23%), VIM+OXA (4.65%), and KPC+NDM (2.79%). Among TCP strains, *A. baumannii* and *K. pneumoniae* were the most prevalent species, accounting for 49.06% (52/106) and 40.57% (43/106) of the TCP strains, respectively. The leading carbapenemase combinations for TCP *A. baumannii* were NDM+OXA+OXA (80.77%) and GES+OXA+OXA (15.38%), while for TCP *K. pneumoniae*, they were KPC+VIM+OXA (39.53%), KPC+NDM+OXA (20.93%), NDM+VIM+OXA (18.60%), and KPC+NDM+VIM (18.60%).

## Patient outcomes profiling with species and combinations of carbapenemases, as well as varying infection routes caused by RCP bacteria

Specific species, different carbapenemase combinations, and the route of infection are crucial for tailoring appropriate treatment strategies and affecting patient outcomes. Different species and combinations of carbapenemases, as well as varying infection routes, may have a significant impact on patient outcomes. To further understand the possible clinical impact on infectious-related issues, the specimen sources and patient outcome information of 160 RCP organisms were collected and visualized (Fig. 3A). Among the RCP bacteria, the leading species and carbapenemase combinations associated with patient death were *K. pneumoniae* carrying NDM+OXA,

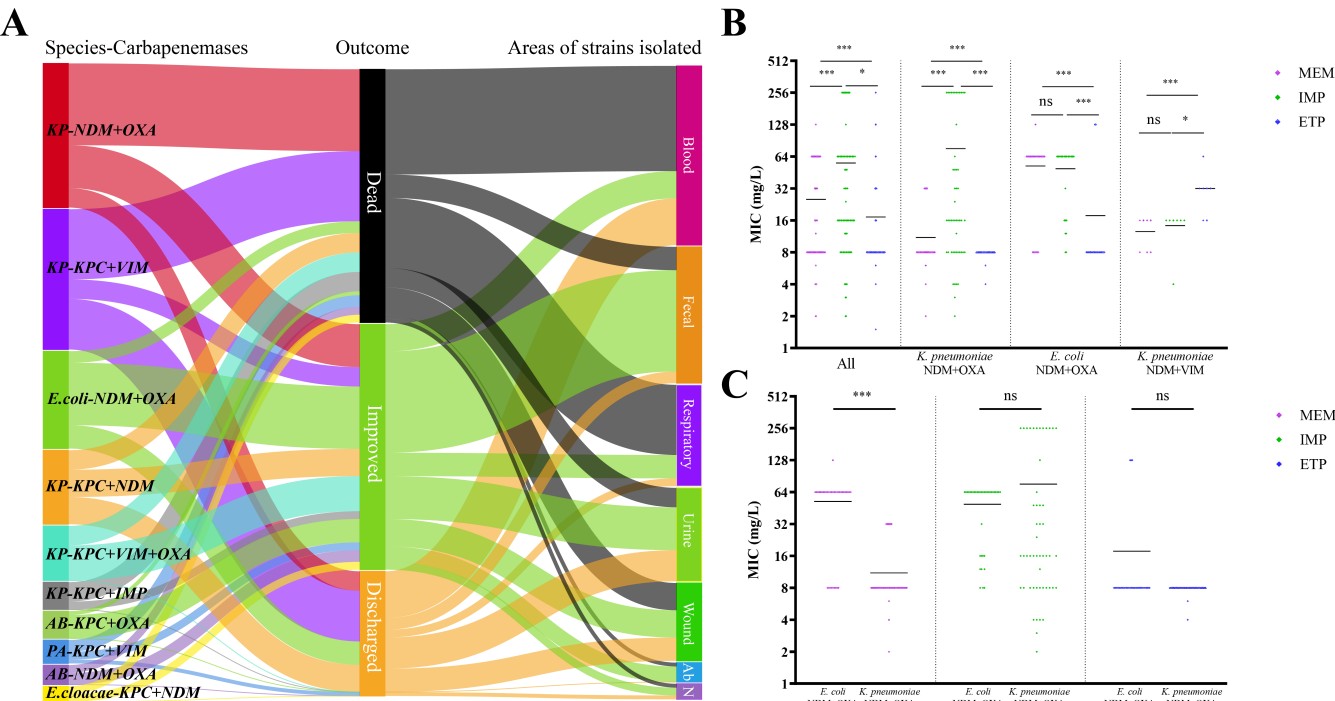

**FIG 3** Clinical impact of RCP strains. (A) Patient outcomes of infections caused by RCP strains. (B) Comparison of minimum inhibitory concentration (MIC) for different carbapenems in NDM+OXA-producing strains. (C) Comparison of carbapenem MICs in NDM+OXA-carrying strains from various species.

KPC+VIM, KPC+NDM, and KPC+IMP. Notably, patients infected with *K. pneumoniae* strains harboring NDM+OXA and KPC+VIM demonstrated mortality rates of 56.76% and 50.00%, respectively. In contrast, patients infected with KPC+NDM-producing *K. pneumoniae* exhibited a mortality rate of 26.32%, while those infected with NDM+OXA-producing *E. coli* showed a mortality rate of 12.00%.

In terms of specimen sources, strains isolated from the bloodstream and respiratory samples constituted approximately 70% of the fatalities. The mortality rates of patients with RCP strains detected from the respiratory tract, blood, wounds, urine, and feces were 69.23%, 58.70%, 35.00%, 20.83%, and 17.14%, respectively.

Additionally, considering the direct involvement of carbapenemases in conferring resistance to carbapenem antibiotics, we conducted a statistical analysis of the minimum inhibitory concentrations (MICs) of RCP strains against carbapenems such as meropenem, imipenem, and ertapenem. We aimed to assess the impact of carbapenemase combinations and species on carbapenem resistance. We excluded strains with approximate MIC values and focused on those with exact MIC values for all three antibiotics. The analysis revealed varying levels of resistance to different carbapenem antibiotics among strains producing carbapenemase combinations (Fig. 3B). For example, NDM+OXA-producing *K. pneumoniae* exhibited a significantly higher MIC for imipenem compared to meropenem, and a higher MIC for meropenem compared to ertapenem. NDM+OXA-producing *E. coli* showed a lower MIC for ertapenem compared to imipenem and meropenem, while NDM+VIM-producing *K. pneumoniae* exhibited a higher MIC for ertapenem compared to imipenem and meropenem. Furthermore, different species carrying the same carbapenemases also displayed variations in resistance to specific carbapenem antibiotics (Fig. 3C). NDM+OXA-producing *E. coli* had a significantly higher MIC for meropenem compared to NDM+OXA-producing *K. pneumoniae*, while their MIC values for imipenem and ertapenem showed no significant difference.

These findings provide valuable insights into the relationship between species, carbapenemase combinations, infection routes, and patient outcomes. They highlight

the importance of considering these factors when designing treatment strategies for infections caused by RCP bacteria.

## Possible antimicrobial resistance and virulence benefits of bacteria producing redundant carbapenemases

A total of 47 DCP strains carrying carbapenemases on separate plasmids, with fully assembled plasmid sequences, were analyzed to investigate the evolutionary advantages of DCP strains. The analysis primarily focused on the carried resistance phenotypes and virulence factors (refer to Fig. 4). The DCP bacterial strain exhibited various carbapenemase combination patterns, including 23 strains with KPC+NDM, 19 strains with NDM+OXA, 3 strains with KPC+OXA, 1 strain with KPC+VIM, and 1 strain with NDM+VIM. To simulate strains carrying a single carbapenemase, the carbapenemase

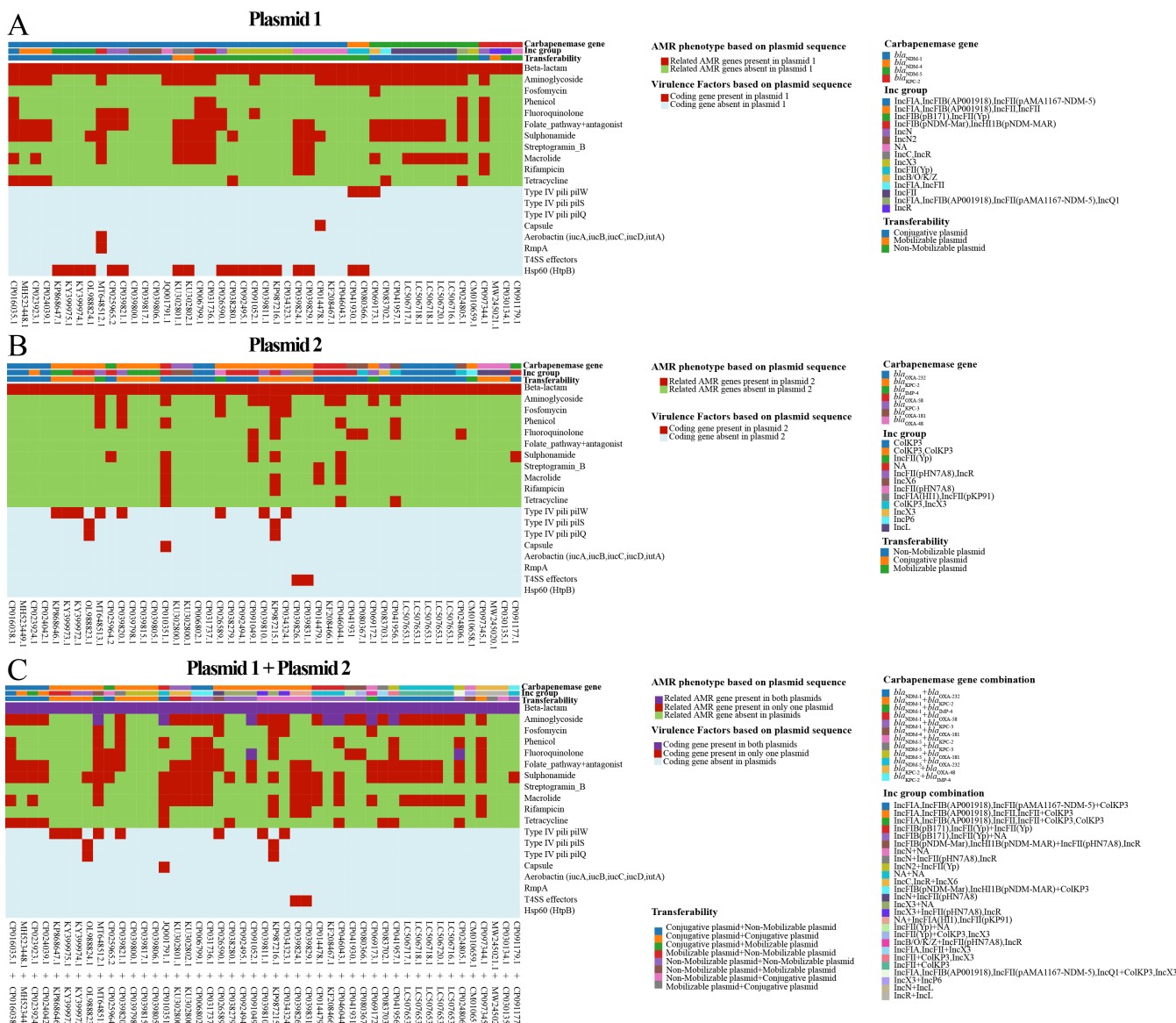

**FIG 4** Bacterial antimicrobial resistance (AMR) and virulence benefits of producing pairs of carbapenemase-carrying plasmids in one strain. (A) The acquired resistant patterns for standard antibiotic classes and the presence of virulence factors carried by the first carbapenemase-carrying plasmid. (B) The acquired resistant patterns for standard antibiotic classes and the presence of virulence factors carried by the second carbapenemase-carrying plasmid. (C) The redundancy and expansion of antibiotic-resistant patterns and virulence factors resulting from combinations of the paired plasmids. Note: in this context, "conjugative" refers to self-transmissible plasmids, while "mobilizable" refers to plasmids that are mobilizable but not self-transmissible.

gene-containing plasmids were divided into two groups: "Plasmid 1" and "Plasmid 2." "Plasmid 1" consisted of 43 plasmids carrying $bla_{NDM}$ and 4 plasmids carrying $bla_{KPC}$, while "Plasmid 2" included 23 $bla_{KPC}$, 22 $bla_{OXA}$, and 2 $bla_{IMP}$ genes.

The overlapping functionality and resistance profiles of carbapenemases suggest redundant functions in DCP strains. We hypothesized that such strains may harbor multiple plasmids carrying carbapenemase genes to extend their resistance profiles and adaptability. These plasmids often carry additional antibiotic resistance genes and virulence factors coding genes. The VRprofile2 predictions of antimicrobial resistance (AMR) genes carried by the plasmids categorized AMR phenotypes into 11 antibiotic classes, including beta-lactam, aminoglycoside, fosfomycin, phenicol, fluoroquinolone, folate pathway antagonist, sulfonamide, streptogramin b, macrolide, rifampicin, and tetracycline. Virulence factors predicted by VFanalyzer and VRprofile2 based on plasmid sequences were also visualized.

By assessing the overlap of antibiotic resistance phenotypes and virulence factors from "Plasmid 1" and "Plasmid 2," the benefits of DCP strains acquiring two carbapene-mase-carrying plasmids were evaluated. "Plasmid 1" exhibited 236 AMR phenotypes, while "Plasmid 2" exhibited 169 AMR phenotypes. Excluding beta-lactam resistance-associated phenotypes, there were 152 non-overlapping antibiotic resistance pheno-types and 9 overlapping ones. The coexistence of plasmid combinations increased the resistance rates of aminoglycosides (80.85%), folate pathway antagonists (51.06%), and sulfonamides (61.17%), surpassing the rates observed in hypothetical single plasmid-carrying strains. Interestingly, all 39 virulence factors did not overlap between the paired carbapenemase-carrying plasmids within the same bacterial strain. This suggests that such plasmids preferentially carry non-overlapping resistance genes and different virulence factors, enhancing the overall antibiotic resistance profile and expanding the pathogenic potential of the strains.

Transferability prediction was also conducted to evaluate the transferability of the plasmids between different bacterial strains or cells, revealed that 12 DCP strains contained two mobilizable (conjugative and mobilizable) plasmids, 12 DCP strains contained two non-mobilizable plasmids, and 23 DCP strains contained one mobilizable and one non-mobilizable plasmid. Among the plasmid combinations, IncFII and ColKP3 were the most frequent, found in 5 DCP strains.

## DISCUSSION

Our study findings show that RCP bacteria are found worldwide and represent a significant global health threat. They have been reported across multiple continents, including Asia, Europe, North America, South America, and Africa. Some countries reported over 100 RCP strains, including China, India, Iran, Thailand, South Korea, Turkey, and Egypt, which appear to be countries with higher population densities.

The earliest article reported three RCP *K. pneumoniae* strains coproducing KPC-2 and VIM-1 carbapenemases in 2009 (13). The earliest isolated RCP strains might be an OXA-181 and VIM-5 coproducing *K. pneumoniae* strain from India in 2007 (14). Temporal analysis reveals a rapid increase in the number of RCP strains, with an especially notable rise in the annual reporting count, reaching approximately 200 cases in recent years. Due to the absence of precise isolation times in many articles or only having a rough range, this study utilized the publication time of the articles as a proxy for the reporting curve of the scientific research community regarding the isolation time of RCP bacteria (also listed in Table S2 if available). Taking into account the time delay in reporting scientific articles, it is possible that the current situation could be even more severe than what has been reported. The worldwide distribution and rapid increase of RCP bacteria underscores the need for international cooperation and collaboration to combat antimicrobial resistance.

The distribution of species-carbapenemase combinations varies significantly, potentially due to differences in the prevalence of bacterial strains carrying single carbapenemase and variations in the acquisition capacity of different species for plasmids. *K. pneumoniae*, *A. baumannii*, and *E. coli* are the predominant species in RCP

strains. This is likely associated with the high clinical prevalence of carbapenem-resistant *K. pneumoniae* (15) and *A. baumannii* (16), as well as *E. coli* (17) being the most frequently isolated Gram-negative bacilli in clinical settings. NDM (18), OXA (19), and KPC (20) are the leading carbapenemases presented in carbapenemase combinations, which is also consistent with their high prevalence among all carbapenem-resistant bacteria.

The RCP bacteria can cause severe and potentially life-threatening infections (21, 22). Various factors, including the species of the bacterial strain, combinations of carbapenemases, and the type of specimen, can influence clinical outcomes. The presence of specific carbapenemases, such as KPC, NDM, IMP, or VIM, in different species can result in variations in the level of drug resistance and influence treatment options (23). DCH strains exhibit varying resistance levels to different carbapenem antibiotics. These variations may be explained by the diverse hydrolytic activities of KPC (24), NDM (25), and OXA (26) carbapenemases toward distinct carbapenem antibiotics, leading to variability in resistance levels. Furthermore, it has been observed that different species carrying the same carbapenemases (NDM+OXA) may display differences in resistance levels to specific carbapenem antibiotics. This observation suggests that certain bacterial species may possess intrinsic resistance mechanisms that further impact their susceptibility to specific carbapenem antibiotics. This highlights the significance of developing personalized treatment approaches for infections caused by various combinations of carbapenemases. Our findings also suggest that the site of infection, as indicated by the specimen type of RCP bacteria isolation, can significantly impact the severity of the infection and patient prognosis.

Regarding the evolutionary dynamics of carbapenemase-carrying strains, there might be three possible reasons. (i) Direct evolutionary drive: the higher usage of carbapenems in clinical settings can create selective pressure on bacteria to acquire more carbapenemases. This allows them to enhance their MIC against carbapenem antibiotics, making them more resistant. (ii) Non-carbapenem evolutionary benefits: carbapenemase-carrying strains may have additional advantages beyond resistance to carbapenem antibiotics. These benefits could contribute to their increased prevalence and clinical adaptability, even in the absence of direct carbapenem usage. (iii) Lack of evolutionary drive: in some cases, the increased prevalence of carbapenemase-carrying strains may not be solely driven by a specific evolutionary force. Instead, it could be a result of stochastic events, such as genetic mutations or horizontal gene transfer, leading to the emergence of multiple carbapenemases without a clear selective advantage.

In this study, we find that the presence of multiple carbapenemases probably increases the likelihood of co-resistance to other classes of antibiotics and the presence of more virulence factors, further complicating the clinical management of infections. Carbapenemase-carrying bacteria are significant because they display high levels of resistance to carbapenems, which are often the drug of choice for infections caused by multidrug resistance strains. Other β-lactam antibiotics may also be ineffective in the presence of carbapenemases. In many cases, combination therapy with multiple antibiotics, including aminoglycosides (27), fluoroquinolones, tetracyclines, colistin (28), ceftazidime/avibactam, and tigecycline may be the last choice. However, our findings suggest that this approach is limited in its effectiveness to RCP strains due to a broader antibiotic-resistant profile, and may even drive carbapenemase-producing strains to acquire more carbapenemases, thus becoming more resistant. One previous meta-analysis also suggests monotherapy or combination therapy for infections due to CROs show no difference in mortality and microbiological cure rates (29). Effective infection control measures, including monitoring the dissemination of resilient strains, innovating novel antibiotics, enhancing infection control protocols, and advocating for the judicious use of antibiotics should be taken to impede the continued emergence and transmission of these perilous pathogens.

The limitations of this study are as follows: firstly, epidemiological analysis may suffer from potential biases in both time and space. The presentation of epidemiological trends relies on the reported time rather than the isolation time of strains, potentially distorting

the depiction of actual trends in strain emergence and development. Furthermore, the increased reporting of strains may also be attributed to the broader application of sequencing technologies, thus the trend analysis primarily serves to highlight the possibility of an increasing trend in the emergence of multi-carbapenemase-producing strains, aimed at drawing attention from experts in the field. We also cannot rule out biases in attention and reporting toward specific species in different regions. Secondly, in the analysis and discussion of clinical outcomes associated with multi-carbapenemase-producing strains, other clinical factors such as variations in treatment regimens, patient demographics including age, gender, and underlying conditions, as well as disparities in healthcare standards across different regions, were not extensively explored. A more rigorous evaluation of clinical outcomes associated with multi-carbapenemase-producing strains necessitates further clinical trials. Thirdly, when examining carbapenem resistance among multi-carbapenemase-producing strains, the removal of ambiguous and uncertain MIC values was implemented to alleviate biases stemming from differing antimicrobial susceptibility testing methodologies and antibiotic concentration ranges across studies. This action resulted in a decreased number of strains included in the analysis, potentially compromising the result's generalizability. The authors did not analyze potential mechanisms influencing the MIC, such as copy number and expression diversity among different species and enzymes due to the complexity of these mechanisms. Lastly, in exploring potential evolutionary mechanisms and advantages conferred by multi-carbapenemase production, the analysis only considered scenarios where carbapenemase genes are carried by plasmids individually, overlooking situations where these genes are carried by the bacterial chromosomes or where single plasmids carry multiple carbapenemase genes, both of which could have implications for strain evolution and adaptability.

## Conclusions

RCP bacteria are widely distributed globally, with an increasing trend over time. Due to differences in the infection routes and the infecting pathogenic species, these bacteria can cause severe clinical infections, even leading to death, as the number of viable treatment options becomes limited in such cases. The presence of multiple carbapenemases increases the chances of co-resistance to other classes of antibiotics and more virulence factors, further complicating the clinical management of infections.

## MATERIALS AND METHODS

### Search strategy and selection criteria

We did a comprehensive literature search using the PubMed database to assess the emergence of RCP bacteria. The timeframe of the searched publications is specified as up to September 2022. The search terms are "double carbapenemases," "multiple carbapenemases," "co-occurrence AND carbapenemase," "co-producing AND carbapenemase," "co-producing AND carbapenemase," "coexistence AND carbapenemase," "KPC AND NDM," "KPC AND OXA," "KPC AND VIM," "KPC AND IMP," "KPC AND GES," "NDM AND OXA," "NDM AND VIM," "NDM AND IMP," "NDM AND GES," "OXA AND VIM," "OXA AND IMP," "OXA AND GES," "VIM AND IMP," "VIM AND GES," and "IMP AND GES" (Table S1). We conducted a thorough examination of the complete texts and reviewed the references cited in the retrieved articles to ensure the inclusion of references pertinent to RCP strains. No study type or language restriction was applied. The criteria to define carbapenemases were KPC, NDM, IMP, VIM, GES, OXA-23, OXA-51, OXA-48, etc. (listed in Table S1). After reviewing the literature, all articles that provided evidence of carbapenemase existence through PCR sequencing, conjugation transfer, or second or third-generation sequencing were included in the study. Articles that did not report RCP strains and those that only mentioned the detection of RCP strains without specific information about the combination of carbapenemase and species were excluded. Based on the

strain names, isolation locations, isolation times, sequence accessions, corresponding patient genders, ages, and other information, duplicate strains were removed.

## Data collection and analysis procedure

A total of 277 articles that meet the inclusion criteria are listed in Table S2. The following data were extracted from the included articles: strain identification, carbapenemase types, geographic distribution, isolation date, antibiotic susceptibility profiles, and sample sources of drug-resistant strains carrying multiple carbapenemases. Clinic-relevant information including the diagnosis, medication, and outcomes of patients has also been collected (Table S2). The epidemiology, clinical significance, and bacterial benefits of RCP strains were analyzed based on the collected information. The flow diagram of the analysis procedure, and inclusion protocols used are given in Fig. S1 in the supplemental material.

## Epidemiology analysis

Articles reporting carbapenemase combinations, and host bacteria species were included for epidemiology characterization of the RCP strains. The country and regional information of the strains is based on the location of the first author's affiliation unless there is a conflict with the information explicitly stated in the text. Since the isolation time of some strains is not specified, to observe the developmental trends of the strains, the publication date of the article will be used as the reporting time to conduct temporal trend analysis, the isolation date was also listed in Table S2.

## Clinical impact analysis

Studies reporting clinically isolated RCP strains along with the diagnosis, treatment, and outcomes of the patient host were included for assessment of the clinical impact of the strains. Sample types, and patient outcomes, have been extracted to evaluate the common infection routes and clinical mortality rates.

Studies reporting carbapenem antibiotic susceptibility of clinically isolated RCP strains were included to assess the possible impact on antibiotic resistance preference and drug availability. All ambiguous MIC values including those with "≥," ">," "<," or "≤" were excluded. Only the MIC values of carbapenem antibiotics with clear numerical values were retained. Subsequently, we selected a subset with a quantity greater than 30 strains for a comparative analysis of carbapenem antibiotic susceptibility. MIC comparison of different carbapenems for NDM+OXA carrying strains was performed using two-tailed paired $t$-tests. Carbapenems MIC comparison of NDM+OXA carrying strains from different species was performed using two-tailed non-paired $t$-tests.

## Antimicrobial resistance and virulence factor analysis and benefit evaluation

The 47 RCP strains carrying two different carbapenemases on separate plasmids and possessing complete plasmid sequencing data were included for genetic analysis (Table S3). The antimicrobial resistance phenotypes and virulence factors were predicted based on plasmid sequences using VRprofile2 (30) and VFanalyzer (31). Clinical diagnosis and breakpoints were extracted as reported by the included studies. The antibiotic class was stratified according to the third level of the WHO Anatomical Therapeutic Chemical Classification System.

## ACKNOWLEDGMENTS

The research was supported by research grants from the Natural Science Foundation of China (No. 81974318 and No. 82302588), China Primary Health Care Foundation (No. MTP2022D027), and Guangdong Province Science and Technology Innovation Strategy Special Fund (No. 2019B020209001).

P.-B.Y. and D.-Q.C. designed the experiments. L.-T.D. and Q.-K.Z. performed the experiments. P.-B.Y., L.-T.D., Q.-K.Z., Y.-X.Z., L.Y., and W.-T.L. analyzed the data. P.-B.Y. and D.-Q.C. wrote the manuscript. All authors read and approved the final manuscript.

The funder had no role in the study design, data collection, data analysis, data interpretation, writing of the manuscript, or the decision to submit.

## AUTHOR AFFILIATIONS

[1]Microbiome Medicine Center, Department of Laboratory Medicine, Zhujiang Hospital, Southern Medical University, Guangzhou, China

[2]Department of Clinical Laboratory, National Center for Respiratory Medicine, National Clinical Research Center for Respiratory Disease, State Key Laboratory of Respiratory Disease, Guangzhou Institute of Respiratory Health, The First Affiliated Hospital of Guangzhou Medical University, Guangzhou, Guangdong, China

## AUTHOR ORCIDs

Ding-Qiang Chen ⓘ http://orcid.org/0000-0001-8717-1651

## FUNDING

| Funder | Grant(s) | Author(s) |
|---|---|---|
| MOST \| National Natural Science Foundation of China (NSFC) | 81974318 | Ding-Qiang Chen |
| MOST \| National Natural Science Foundation of China (NSFC) | 82302588 | Pei-Bo Yuan |
| China Primary Health Care Foundation (CPHCF) | MTP2022D027 | Ding-Qiang Chen |
| GDSTC \| Special Fund Project for Science and Technology Innovation Strategy of Guangdong Province (Guangdong Province Science and Technology Innovation Strategy Special Fund Project) | 2019B020209001 | Ding-Qiang Chen |

## DATA AVAILABILITY

All materials are available from the corresponding author.

## ADDITIONAL FILES

The following material is available online.

### Supplemental Material

**Supplemental material (Spectrum00008-24-s0001.tif).** Fig. S1.
**Table S1 (Spectrum00008-24-s0002.xlsx).** Carbapenemase subtypes present.
**Table S2 (Spectrum00008-24-s0003.xlsx).** Characteristics of 1,780 redundant carbapenemase-producing strains.
**Table S3 (Spectrum00008-24-s0004.xlsx).** Double carbapenemase-producing strains with fully assembled plasmid sequences.

### Open Peer Review

**PEER REVIEW HISTORY (review-history.pdf).** An accounting of the reviewer comments and feedback.

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
