## [Reviewer comments · Microbiology Spectrum]

Microbiology Spectrum

Global Emergence of Double and Multi-Carbapenemase Producing Organisms: Epidemiology, Clinical Significance, and Evolutionary Benefits on AMR and Virulence

Pei-bo Yuan, Li-Ting Dai, Qi-Ke Zhang, Yu-Xia Zhong, Wanting Liu, Ling Yang, and Dingqiang Chen

Corresponding Author(s): Dingqiang Chen, Department of Laboratory Medicine, Zhujiang Hospital, Southern Medical University

Review Timeline:

Submission Date:	January 4, 2024
Editorial Decision:	February 16, 2024
Revision Received:	April 15, 2024
Accepted:	May 2, 2024

Editor: Vittal Ponraj

Reviewer(s): The reviewers have opted to remain anonymous.

Transaction Report:

DOI: <https://doi.org/10.1128/spectrum.00008-24>

Re: Spectrum00008-24 (Global Emergence of Double and Multi-Carbapenemase Producing Organisms: Epidemiology, Clinical Significance, and Evolutionary Benefits on AMR and Virulence)

Dear Dr. Dingqiang Chen:

Thank you for the privilege of reviewing your work. Below you will find my comments, instructions from the Spectrum editorial office, and the reviewer comments. We would be willing to consider a revised version of this paper with the comments/queries from the reviewers appropriately addressed.

Revision Guidelines

Sincerely,

Vittal Ponraj Ph.D., SM(ASCP)CM
Editor
Microbiology Spectrum

Reviewer #1 (Comments for the Author):

Lines 78/79 - Please add citation(s)

Line 83 - Add citation for IDSA guideline - <https://www.idsociety.org/practice-guideline/amr-guidance/>

Lines 87-89 - Please add citation(s)

Lines 89-91 - Please clarify that aztreonam is active against isolated NDM producers, but that organisms often have multiple mechanisms of resistance present (e.g., ESBL, other carbapenemases, porin/efflux mutations) that will render aztreonam inactive.

Lines 97-98 - please add citation supporting prolonged inpatient stays, healthcare cost and increased morbidity/mortality

Lines 107-129 - Please address in your discussion if any of the geographic or species distribution differences could be explained by reporting biases (e.g., a study on African isolates that was solely focused on *Acinetobacter*).

Lines 279-281 - Please comment on the potential impact of more widely available and affordable sequencing technologies could have impacted the reporting trends in you analysis

Lines 300-305, 336-340 - Please comment on the impact variable antimicrobial treatment choices could have on your clinical outcome data, especially in light of variability in access to novel beta-lactam/beta-lactamase inhibitor combination agents (ceftazidime/avibactam, meropenem/vaborbactam, sulbactam/durlobactam) and cefiderocol.

Line 349 - Your discussion must include content related to the limitations of your study

Line 358 - Please add information regarding what efforts your team undertook to prevent inclusion of duplicate data (e.g., multiple publications reporting data on the same or overlapping isolate sets).

Lines 359-366 - Please address how your search strategy addressed finding dual/multiple carbapenemase producing isolates that were included in publications that either were not focused on this issue or did not identify they contained this data in the abstract

Lines 363-366 - Please clarify why GES was not included in the search terms above but is noted as part of the carbapenemase definition?

Line 368 - Please address your decision to include OXA-51 which is a chromosomal enzyme in *Acinetobacter* vs the other predominantly plasmid mediated enzymes?

Lines 390-393 - Please justify the use of this strategy vs excluding isolates without a clear isolation date? It is easy to imagine articles you included could have potentially included 5-10+ years of isolates and with your strategy the isolation dates would be significantly mis-categorized.

Lines 401-408 - This approach is not appropriate for demonstrating the impact of multiple carbapenemase enzymes in varying organisms. A more appropriate analysis here would require reporting of all MIC values for organisms expressing single and multiple enzymes of interest. Your analysis approach also does not address the potential impact of enzyme copy number on phenotypic MIC values. Given the available retrospective data, your publication should likely restrict itself to a simple descriptive and non-comparative presentation of the MIC data.

Reviewer #2 (Comments for the Author):

Summary of Key Findings

Thank you for the opportunity to review this work. In this manuscript, Yuan et al. performed a retrospective analysis to establish the prevalence of redundant carbapenemase producing (RCP) bacteria, and used sequencing data to evaluate their clinical impact and virulence factors. Key findings include a summary of 1,780 reported RCPs globally, with over 93% harboring two carbapenemases. Reports of up to six carbapenemases have been reported from different Enterobacterales isolates, and the incidence of these reports have increased over time. It is possible that infection with strains harboring multiple carbapenemases may be associated with more severe disease, as well a increased risk of co-resistance.

Major Concerns

1. In the search strategy section, the timeframe of searched publications is not provided. Please include this.
2. Lines 401-403: I am concerned that the exclusion of ambiguous MIC values (e.g., those that are non-discrete) results in the exclusion of important information. However, this is difficult to ascertain when the interpretive criteria (CLSI, EUCAST)/breakpoints used is unknown. For example, any MIC that came from an automated system and uses current CLSI breakpoints with have a resistant meropenem MIC of ≥ 4 . While comparing MICs is not necessarily the point of this study, I think further explanation regarding why discrete MICs were the only included (and what type of information this leaves out) would be helpful.
3. Lines 179-184: The interpretation of these data are challenging given that other clinical context is not available. Was analysis

done to account for other factors that may be associated with death? If not, perhaps this can simply be re-worded to say that patients who died more commonly were infected with strains harboring "xyz" resistance mechanism.

4. I do not see a limitations section in this paper, and suggest that one be included. Topics to note in this section include those mentioned above, the inability to determine whether the prevalence of mechanisms has been increasing over time or technology has become more able to detect these (or both), any challenges related to being unable to clearly say the infection caused death (elaborate on how clinical information was adjudicated or how analyses were done).

Minor Concerns

1. Lines 378-379: the authors note that antibiotic susceptibility profiles were included in the data that were extracted from published studies. It is important to note in the limitations that there is significant variability in the way laboratories perform susceptibility testing, including the use of interpretive breakpoints, which directly impacts the way minimum inhibitory concentrations are interpreted.
2. Line 34: The text states that this study aims to analyze the epidemiology and genetic mechanisms of RCP strains with the goal of facilitating targeted surveillance and control measures. I feel that it might be more appropriate to say something like "support targeted surveillance and control measures." Unless there is a pipeline in place that can support the implementation of actions this study recommends, I suggest making this change.
3. Throughout the manuscript, antimicrobial is hyphenated and should not be.
4. Line 42, should be "100 RCP strains"
5. Line 48, please provide a citation for the mortality rates mentioned
6. Lines 96-98, while it is well known that AMR contributes to poor clinical outcomes, I think that a citation is needed here
7. Lines 147-148: consider re-wording to something like, "Consider re-wording to something like: "to investigate the association between different carbapenemase combinations and bacterial species."
8. Line 185-186: consider re-wording, specimens and sources are not responsible for death.
9. Lines 230-232: Is this hypothesis accepted in the scientific community? Or is this a hypothesis the authors are making? Please clarify. If this is something that has been published on, please provide a citation.
10. Lines 314-316 state, "Our findings also indicate that the specimen type of RCP bacteria isolation can significantly influence the severity of the infection and patient prognosis." Consider re-wording. I think what is meant here is that the severity of infection differs by site of infection (as indicated by where the specimens came from, or specimen type).

While we are willing to consider a revised version of this paper at Spectrum, it would be in your best interest to improve the writing. I recommend that you ask a colleague of yours who is a native English speaker to read and provide you some feedback on the writing. You are also welcome to use one of the services here: <https://journals.asm.org/content/language-editing-services>

Global 1 Emergence of Double and Multi-Carbapenemase Producing Organisms: Epidemiology, Clinical Significance, and Evolutionary Benefits on AMR and Virulence.

Summary of Key Findings (200-250 words)

Thank you for the opportunity to review this work. In this manuscript, Yuan et al. performed a retrospective analysis to establish the prevalence of redundant carbapenemase producing (RCP) bacteria, and used sequencing data to evaluate their clinical impact and virulence factors. Key findings include a summary of 1,780 reported RCPs globally, with over 93% harboring two carbapenemases. Reports of up to six carbapenemases have been reported from different Enterobacterales isolates, and the incidence of these reports have increased over time. It is possible that infection with strains harboring multiple carbapenemases may be associated with more severe disease, as well a increased risk of co-resistance.

Major Concerns (at most 5-6):

1. In the search strategy section, the timeframe of searched publications is not provided. Please include this.
2. Lines 401-403: I am concerned that the exclusion of ambiguous MIC values (e.g., those that are non-discrete) results in the exclusion of important information. However, this is difficult to ascertain when the interpretive criteria (CLSI, EUCAST)/breakpoints used is unknown. For example, any MIC that came from an automated system and uses current CLSI breakpoints with have a resistant meropenem MIC of ≥ 4 . While comparing MICs is not necessarily the point of this study, I think further explanation regarding why discrete MICs were the only included (and what type of information this leaves out) would be helpful.
3. Lines 179-184: The interpretation of these data are challenging given that other clinical context is not available. Was analysis done to account for other factors that may be associated with death? If not, perhaps this can simply be re-worded to say that patients who died more commonly were infected with strains harboring “xyz” resistance mechanism.
4. I do not see a limitations section in this paper, and suggest that on be included. Topics to note in this section include those mentioned above, the inability to determine whether the prevalence of mechanisms has been increasing over time or technology has become more able to detect these (or both), any challenges related to being unable to clearly say the infection caused death (elaborate on how clinical information was adjudicated or how analyses were done).

Minor Concerns (at most 5-20 in bullet points):

1. Lines 378-379: the authors note that antibiotic susceptibility profiles were included in the data that were extracted from published studies. It is important to note in the limitations that there is significant variability in the way laboratories perform susceptibility testing, including the use of interpretive breakpoints, which directly impacts the way minimum inhibitory concentrations are interpreted.

2. Line 34: The text states that this study aims to analyze the epidemiology and genetic mechanisms of RCP strains with the goal of facilitating targeted surveillance and control measures. I feel that it might be more appropriate to say something like “support targeted surveillance and control measures.” Unless there is a pipeline in place that can support the implementation of actions this study recommends, I suggest making this change.
3. Throughout the manuscript, antimicrobial is hyphenated and should not be.
4. Line 42, should be “100 RCP strains”
5. Line 48, please provide a citation for the mortality rates mentioned
6. Lines 96-98, while it is well known that AMR contributes to poor clinical outcomes, I think that a citation is needed here
7. Lines 147-148: consider re-wording to something like, “Consider re-wording to something like: “to investigate the association between different carbapenemase combinations and bacterial species.”
8. Line 185-186: consider re-wording, specimens and sources are not responsible for death.
9. Lines 230-232: Is this hypothesis accepted in the scientific community? Or is this a hypothesis the authors are making? Please clarify. If this is something that has been published on, please provide a citation.
10. Lines 314-316 state, “Our findings also indicate that the specimen type of RCP bacteria isolation can significantly influence the severity of the infection and patient prognosis.” Consider re-wording. I think what is meant here is that the severity of infection differs by site of infection (as indicated by where the specimens came from, or specimen type).

We would like to thank the reviewers for taking the time to review our manuscript, and help us to improve our work. Below are our responses to the reviewer's suggestions, with the replied text highlighted in blue. The modifications to the main text are all highlighted in yellow in the Marked-up manuscript, making it convenient for reviewers to quickly locate them.

Reviewer #1 (Comments for the Author):

Lines 78/79 - Please add citation(s)

Response: Thanks for reminding and we have added the citation as reference 6 to the revised manuscript (Line 79 in text and lines 490-491 in references).

Line 83 - Add citation for IDSA guideline - <https://www.idsociety.org/practice-guideline/amr-guidance/>

Response: Thank you for the reminder. We have added the citation to the revised manuscript as reference 8 (Line 83 in text and lines 494-497 in references).

Lines 87-89 - Please add citation(s)

Response: Thank you for the advice, we have added the citation to the revised manuscript as reference 9 (Line 88 in text and lines 498-500 in references).

Lines 89-91 - Please clarify that aztreonam is active against isolated NDM producers, but that organisms often have multiple mechanisms of resistance present (e.g., ESBL, other carbapenemases, porin/efflux mutations) that will render aztreonam inactive.

Response: Thank you for your kindly comment. We have revised the manuscript and provided clarification on this point. (Line 89-92).

Lines 97-98 - please add citation supporting prolonged inpatient stays, healthcare cost and increased morbidity/mortality

Response: Thank you for your kindly reminder. After reviewing the literature, we found no reliable direct comparisons between hospitalization time and mortality rates for multi-carbapenemase-producing strains and single-carbapenemase-producing strains. The author's speculative statement about the impact of reducing drug selection to alleviate treatment challenges has been removed in response to the reviewers' reminder.

Lines 107-129 - Please address in your discussion if any of the geographic or species distribution differences could be explained by reporting biases (e.g., a study on African isolates that was solely focused on *Acinetobacter*).

Response: Thank you for the valuable suggestion. We had indeed observed some geographic related species distribution differences (Fig1.B), after analyzing the sources of the reported strains, we believe that this is not due to a single article reporting a large number of strains. For example, we observed multi-carbapenemases producing *Pseudomonas* were mostly found in Americas (63%,17 out of 27), these *Pseudomonas* strains were reported by 9 different articles (article orders in appendix file 2: 19, 20, 43, 44, 55, 208, 216, 233, 265), and isolated from Chile, Brazil, the United States, and Chile, and so on. We also observed higher proportion of multi-carbapenemases producing *Acinetobacter* strains in Africa compared to other continents (rather than higher number) as mentioned by the reviewer, these strains were reported by 13 articles (article orders in appendix file 2: 35, 56, 92, 93, 94, 150, 153, 175, 219, 232, 253, 261) from 7 countries (Algeria, Egypt, Kenya, Libya, Malaysia, Morocco, and Sudan). While we still cannot rule out biases in attention and reporting towards specific species in different regions (mentioned in the revised manuscript line 350-351), it is highly unlikely that the bias is due to a single article reporting a large number of strains.

Lines 279-281 - Please comment on the potential impact of more widely available and affordable sequencing technologies could have impacted the reporting trends in your analysis.

Response: The authors would like to thank the reviewer for the valuable suggestion. According to the reviewer's suggestion, we have included the potential bias arising from increased reporting due to advancements and widespread adoption of sequencing technologies in our study limitations section line 343-371.

Lines 300-305, 336-340 - Please comment on the impact variable antimicrobial treatment choices could have on your clinical outcome data, especially in light of variability in access to novel beta-lactam/beta-lactamase inhibitor combination agents (ceftazidime/avibactam, meropenem/vaborbactam, sulbactam/durlobactam) and cefiderocol.

Response: Thank you for the valuable suggestion. Due to the inclusion of cases from different countries and regions, this study did not thoroughly discuss the potential impact of variable antimicrobial treatment choices on the treatment outcomes of infections. This is also a limitation of our study. In response to the reviewer's suggestion, the authors have included this point in the limitations section of the manuscript line 343-371.

Line 349 - Your discussion must include content related to the limitations of your study

Response: Thank you very much for this valuable suggestion. The authors have added a section on limitations in the revised manuscript, discussing the shortcomings of the article in the revised manuscript line 343-371.

Line 358 - Please add information regarding what efforts your team undertook to prevent inclusion of duplicate data (e.g., multiple publications reporting data on the same or overlapping isolate sets).

Response: The authors would like to thank the reviewer for the valuable suggestion. The authors have thoroughly reviewed all the included literature and collected detailed information such as the names of the strains mentioned in the literature, the time and location of isolation, the gender, age, diagnosis, treatment, and outcome of the infected patients, and the type of samples from which the strains

were isolated. Additionally, we have collected genomic and plasmid accession numbers for strains with sequencing-related information to avoid duplicate counting. We have checked the detailed isolation information from different publications to insure that overlapping isolate be deleted. We supplemented the methods section with this information in the revised manuscript line 400-402.

Lines 359-366 - Please address how your search strategy addressed finding dual/multiple carbapenemase producing isolates that were included in publications that either were not focused on this issue or did not identify they contained this data in the abstract

Response: Thank you for the valuable suggestion. The authors utilized multiple approaches to include research papers. The authors of the document utilized multiple approaches to include research papers. Firstly, we conducted direct keyword searches to identify relevant literature. Secondly, we thoroughly read the full text of the retrieved literature to identify any additional strains that were referenced or mentioned. We aimed to include a comprehensive range of literature but acknowledge the possibility of minor omissions despite their efforts to minimize them. We supplemented the methods section with this information in the revised manuscript line 391-393.

(1)Lines 363-366 - Please clarify why GES was not included in the search terms above but is noted as part of the carbapenemase definition?

Response: Thanks a lot for the kindly reminder. We had conducted a search for GES, but inadvertently omitted it when listing the search terms. We have corrected this error in the revised manuscript line 388-391 and Appendix File 1.

Line 368 - Please address your decision to include OXA-51 which is a chromosomal enzyme in Acinetobacter vs the other predominantly plasmid mediated enzymes?

Response: The authors would like to thank the reviewer for the valuable suggestion. When including carbapenemases, we did not exclude enzymes carried on

the chromosome, nor did we exclude predominantly plasmid mediated enzymes if they integrated into the chromosome.

Lines 390-393 - Please justify the use of this strategy vs excluding isolates without a clear isolation date? It is easy to imagine articles you included could have potentially included 5-10+ years of isolates and with your strategy the isolation dates would be significantly mis-categorized.

Response: Thank you for the valuable suggestion. We totally agree with the reviewer's comment that the reporting year cannot be equated with the isolation time of strains, nor can it accurately reflect the trend of strain detection. The decision to plot based on reported dates was primarily driven by considerations of data completeness. The main intention of the authors was to highlight the significant increase in the number of reported strains over the years, raising awareness among experts in the field and potentially prompting further clinical research to elucidate the epidemiological and developmental trends of multi-carbapenemase-producing strains. We also collected and listed isolation time information for all available strains in Appendix 2, which can be reached by researchers who might be interested. We also added this to the limitation section in the manuscript line 343-371.

Lines 401-408 - This approach is not appropriate for demonstrating the impact of multiple carbapenemase enzymes in varying organisms. A more appropriate analysis here would require reporting of all MIC values for organisms expressing single and multiple enzymes of interest. Your analysis approach also does not address the potential impact of enzyme copy number on phenotypic MIC values. Given the available retrospective data, your publication should likely restrict itself to a simple descriptive and non-comparative presentation of the MIC data.

Response: Thank you very much for the valuable advice. We agree with the reviewer's suggestion to compare the MICs of strains producing single enzymes versus those producing multiple enzymes for a particular species. This could provide a more direct understanding of whether the presence of multiple carbapenemase genes

in a strain affects the MIC of that species to carbapenems. The main challenge lies in obtaining comparable single-enzyme and multi-enzyme strains with similar genetic backgrounds for comparison. However, we believe that our current analysis of MICs is valuable and can be presented at a quantitative level.

In figure 3.B, we compared the MIC differences of different carbapenems for most prevalent species and enzymes (*K. pneumoniae* NDM+OXA, *E. coli* NDM+OXA, and *K. pneumoniae* NDM+VIM), which might provide some reference value for drug selection in infections caused by multi-carbapenemase-producing strains. In this comparison, strain genetic background variations do not affect the results as the comparison of minimum inhibitory concentrations (MIC) to different antibiotics is self-paired. Additionally, there are no methodological variations as a consistent approach is used to determine MIC values for different antibiotics. The analysis specifically concentrated on combinations with a higher number of strains.

In figure 3.C. we compared the MIC differences of carbapenems between different species strains share the same enzymes (*K. pneumoniae* NDM+OXA and *E. coli* NDM+OXA). *E. coli* and *K. pneumoniae* are the two most clinically isolated Gram-negative bacteria and are high-risk species for clinical infections. Our analysis aims to explore whether NDM+OXA have differential impacts on carbapenem resistance across different species, which is also valuable for guiding treatment selection.

A limitation of the authors here is the lack of in-depth discussion on the underlying reasons for the phenomenon, which could be due to the copy number differences of carbapenemase genes mentioned by the reviewer, the hydrolytic preferences of carbapenemases to different carbapenem antibiotics, and carbapenemase express differences. We would discuss this aspect in the limitations section in line 363-365.

Reviewer #2 (Comments for the Author):

Summary of Key Findings

Thank you for the opportunity to review this work. In this manuscript, Yuan et al. performed a retrospective analysis to establish the prevalence of redundant carbapenemase producing (RCP) bacteria, and used sequencing data to evaluate their clinical impact and virulence factors. Key findings include a summary of 1,780 reported RCPs globally, with over 93% harboring two carbapenemases. Reports of up to six carbapenemases have been reported from different Enterobacterales isolates, and the incidence of these reports have increased over time. It is possible that infection with strains harboring multiple carbapenemases may be associated with more severe disease, as well a increased risk of co-resistance.

Major Concerns

1. In the search strategy section, the timeframe of searched publications is not provided. Please include this.

Response: Thank you very much for the valuable advice. Articles collected by the authors were published up to September 2022. We supplemented the methods section with this information in the revised manuscript line 383-384.

2. Lines 401-403: I am concerned that the exclusion of ambiguous MIC values (e.g., those that are non-discrete) results in the exclusion of important information. However, this is difficult to ascertain when the interpretive criteria (CLSI, EUCAST)/breakpoints used is unknown. For example, any MIC that came from an automated system and uses current CLSI breakpoints with have a resistant meropenem MIC of ≥ 4 . While comparing MICs is not necessarily the point of this study, I think further explanation regarding why discrete MICs were the only included (and what type of information this leaves out) would be helpful.

Response: Thank you for the thorough and thoughtful review. The exclusion of ambiguous MIC values (e.g., those that are non-discrete) in this study's main drawback is the reduction in the number of strains available for MIC comparison. Nevertheless, the authors still believe that this analysis is valuable and provides useful insights for future drug selection.

In Figure 3.B, we compared the MIC differences of different carbapenems for most prevalent species and enzymes (*K. pneumoniae* NDM+OXA, *E. coli* NDM+OXA, and *K. pneumoniae* NDM+VIM), which might provide some reference value for drug selection in infections caused by multi-carbapenemase-producing strains. In this comparison, strain genetic background variations do not affect the results as the comparison of minimum inhibitory concentrations (MIC) to different antibiotics is self-paired. Additionally, there are no methodological variations as a consistent approach is used to determine MIC values for different antibiotics.

In Figure 3.C. we compared the MIC differences of carbapenems between different species strains share the same enzymes (*K. pneumoniae* NDM+OXA and *E. coli* NDM+OXA). *E. coli* and *K. pneumoniae* are the two most clinically isolated Gram-negative bacteria and are high-risk species for clinical infections. Our analysis aims to explore whether NDM+OXA have differential impacts on carbapenem resistance across different species, which is also valuable for guiding treatment selection.

3. Lines 179-184: The interpretation of these data are challenging given that other clinical context is not available. Was analysis done to account for other factors that may be associated with death? If not, perhaps this can simply be re-worded to say that patients who died more commonly were infected with strains harboring "xyz" resistance mechanism.

Response: Thank you very much for the valuable advice. We fully agree with the reviewer's suggestion that many other clinical factors could potentially influence patient outcomes. Accordingly, we have revised the wording in the manuscript line

178-182 as suggested by the reviewer.

4. I do not see a limitations section in this paper, and suggest that one be included. Topics to note in this section include those mentioned above, the inability to determine whether the prevalence of mechanisms has been increasing over time or technology has become more able to detect these (or both), any challenges related to being unable to clearly say the infection caused death (elaborate on how clinical information was adjudicated or how analyses were done).

Response: Thank you very much for the valuable suggestion. Following the reviewer's advice, the authors have added a limitations section in the manuscript line 343-371, which discusses the inability to determine whether the prevalence of mechanisms has been increasing over time or if technology has become more capable of detecting these, as well as not being able to clearly say the infection caused death, and other limitations.

Minor Concerns

1. Lines 378-379: the authors note that antibiotic susceptibility profiles were included in the data that were extracted from published studies. It is important to note in the limitations that there is significant variability in the way laboratories perform susceptibility testing, including the use of interpretive breakpoints, which directly impacts the way minimum inhibitory concentrations are interpreted.

Response: Thank you very much for the valuable suggestion. We have noted this in the limitations in the revised manuscript line 343-371 as suggested by the reviewer.

2. Line 34: The text states that this study aims to analyze the epidemiology and genetic mechanisms of RCP strains with the goal of facilitating targeted surveillance and control measures. I feel that it might be more appropriate to say something like "support targeted surveillance and control measures." Unless there is a pipeline in place that can support the implementation of actions this study recommends, I suggest

making this change.

Response: Thank you very much for the kindly suggestion. We have revised the wording in the manuscript as suggested by the reviewer in line 32-34.

3. Throughout the manuscript, antimicrobial is hyphenated and should not be.

Response: Thank you very much for the kindly reminder. We have revised the wording in the manuscript as suggested by the reviewer.

4. Line 42, should be "100 RCP strains"

Response: Thank you very much for the kindly reminder. We have revised the wording in the manuscript line 42 as suggested.

5. Line 48, please provide a citation for the mortality rates mentioned

Response: Thank you very much for the kindly advise. The mortality rates are based on the statistics from this study and correspond to Figure 3.A. To avoid ambiguity, we had revised the sentence to “In patients with RCP strains isolated from the bloodstream and respiratory system, the mortality rates are 58.70% and 69.23%, respectively” in line 47-48.

6. Lines 96-98, while it is well known that AMR contributes to poor clinical outcomes, I think that a citation is needed here.

Response: Thank you for the advice. Since there are no reliable direct comparisons of clinical outcomes between multi-carbapenemase-producing strains and single-carbapenemase-producing strains, we decide to remove this sentence in response to the reviewers' reminder.

7. Lines 147-148: consider re-wording to something like, "to investigate the association between different carbapenemase combinations and bacterial species."

Response: Thank you very much for the kindly suggestion. We have revised the wording in the manuscript as follows “To investigate the association between different carbapenemase combinations and bacterial species, we examined the distribution of

species and carbapenemase proportions in DCP strains (Fig. 2c) and TCP strains (Fig. 2d).” line 145-147

8. Line 185-186: consider re-wording, specimens and sources are not responsible for death.

Response: Thank you very much for the kindly suggestion. We have revised the wording as follows: “In terms of specimen sources, strains isolated from the bloodstream and respiratory samples constituted approximately 70% of the fatalities.” in line 183-184.

9. Lines 230-232: Is this hypothesis accepted in the scientific community? Or is this a hypothesis the authors are making? Please clarify. If this is something that has been published on, please provide a citation.

Response: Thank you very much for the kindly advise. This is a hypothesis the authors are making, so we change the word “It is hypothesized that” into “we hypothesized that” in line 228.

10. Lines 314-316 state, "Our findings also indicate that the specimen type of RCP bacteria isolation can significantly influence the severity of the infection and patient prognosis." Consider re-wording. I think what is meant here is that the severity of infection differs by site of infection (as indicated by where the specimens came from, or specimen type).

Response: Thank you very much for the kindly and helpful advice. We have revised the wording as follows “Our findings also suggest that the site of infection, as indicated by the specimen type of RCP bacteria isolation, can significantly impact the severity of the infection and patient prognosis.” in the revised manuscript line 309-311.

Re: Spectrum00008-24R1 (Global Emergence of Double and Multi-Carbapenemase Producing Organisms: Epidemiology, Clinical Significance, and Evolutionary Benefits on AMR and Virulence)

Dear Dr. Dingqiang Chen:

Your manuscript has been accepted, and I am forwarding it to the ASM production staff for publication. Your paper will first be checked to make sure all elements meet the technical requirements. ASM staff will contact you if anything needs to be revised before copyediting and production can begin. Otherwise, you will be notified when your proofs are ready to be viewed.

Sincerely,

Vittal Prakash Ponraj Ph.D., SM(ASCP)CM
Editor
Microbiology Spectrum